# *In vivo* monitoring of leukemia-niche interactions in a zebrafish xenograft model

Anja Arner[1], Andreas Ettinger[2], Bradley Wayne Blaser[3], Bettina Schmid[4], Irmela Jeremias[1,5], Nadia Rostam[1,6], Vera Binder-Blaser[1] *

1 Department of Pediatric Hematology/Oncology, Dr. von Hauner Children's Hospital, Ludwig Maximilians University (LMU), Munich, Germany, 2 Institute of Epigenetics and Stem Cells, Helmholtz Centre Munich, German Research Center for Environmental Health, Munich, Germany, 3 Division of Hematology, The Ohio State University Comprehensive Cancer Center, Columbus, Ohio, United States of America, 4 German Center for Neurodegenerative Diseases (DZNE), Munich, Germany, 5 Research Unit Apoptosis in Hematopoietic Stem Cells, Helmholtz Centre Munich, German Research Center for Environmental Health, Munich, Germany, 6 Department of Biology, University of Sulaimani, Sulaymaniyah, Iraq

* vera.binder-blaser@med.uni-muenchen.de

**Data Availability Statement:** All relevant data are within the manuscript and its Supporting Information files.

## Abstract

Acute lymphoblastic leukemia (ALL) is the most common type of malignancy in children. ALL prognosis after initial diagnosis is generally good; however, patients suffering from relapse have a poor outcome. The tumor microenvironment is recognized as an important contributor to relapse, yet the cell-cell interactions involved are complex and difficult to study in traditional experimental models. In the present study, we established an innovative larval zebrafish xenotransplantation model, that allows the analysis of leukemic cells (LCs) within an orthotopic niche using time-lapse microscopic and flow cytometric approaches. LCs homed, engrafted and proliferated within the hematopoietic niche at the time of transplant, the caudal hematopoietic tissue (CHT). A specific dissemination pattern of LCs within the CHT was recorded, as they extravasated over time and formed clusters close to the dorsal aorta. Interactions of LCs with macrophages and endothelial cells could be quantitatively characterized. This zebrafish model will allow the quantitative analysis of LCs in a functional and complex microenvironment, to study mechanisms of niche mediated leukemogenesis, leukemia maintenance and relapse development.

## Introduction

Acute lymphoblastic leukemia (ALL) is the most common malignancy during childhood, with B-cell precursor (BCP)-ALL as the most common type of pediatric leukemia [1]. Prognosis after initial diagnosis is good (10 year survival rates ≥90%), due to the immense progress in diagnostics and treatment over the last four decades [2]. However, for patients suffering from refractory disease or relapse, outcome is still poor (5-year survival rates 50–60%). Current risk classification protocols for ALL exclusively consider cell intrinsic features like cytogenetics and molecular genetics of the leukemic cell [3,4]. Therefore, considering features of the surrounding tissue—the niche—might help to improve treatment options in refractory disease and

**Funding:** V.B-B received funds from the German Jose Carreras Foundation and the Bettina-Bräu Stiftung. The funders did not play any role in the study design, data collection and analysis, decision to publish, or preparation of the manuscript.

**Competing interests:** The authors have declared that no competing interests exist.

prevent relapse development. In the field of stem cell research, the term 'niche' describes the place of residence of stem cells where they can fulfill their desired function. Hence, the niche is (i) the anatomic region where stem cells reside and (ii) a distinct locale where specific signals are available to enable stem cell functions, such as self-renewal and differentiation. These specific signals can be provided by the extracellular matrix, cellular components, and secreted molecules which in turn help define stem cell fate [5–7]. With the development of the cancer stem cell concept in leukemia, which describes a distinct subpopulation of LCs possessing features of stem cells such as self-renewal [8,9], the question arose if specialized niches are equally important to maintain the leukemia, as they are for healthy tissue. In case the niche is decisive for leukemia development or maintenance, its components are very attractive therapeutic targets to eliminate malignant cells [5].

There is growing evidence that in addition to cell-intrinsic changes, the niche plays a pivotal role in leukemogenesis, leukemia maintenance and ultimately in relapse development [10]. However, while more and more is known about the signals that drive the interplay of LCs and the microenvironment [11], further detailed studies are needed to comprehensively understand the niches' functions.

Models to study the complex interplay of several cell types within a functional surrounding are scarce. In 2005, the first xenotransplantation of human cancer cells into zebrafish was performed [12]. Subsequently, further studies of different tumor entities were performed [13,14] and the process of xenotransplantation was optimized. Meanwhile, routine transplantation workflows were successfully applied by many groups and this allowed the efficient injection of several hundred embryos within only a few hours (h). Larval xenografts were an appreciated model that were successfully used to study diverse processes, such as tumor metastasis [15,16] and angiogenesis [13,14,17]. The feasibility to monitor fluorescently labeled cells in the living organism in real time over an extended period of time, due to the embryos' optical transparency [18], makes this model highly interesting. Furthermore, numerous studies emphasize the potential of the larval zebrafish as a preclinical model to test drugs [19–23].

The hematopoietic niches in zebrafish are well characterized and it could be shown that hematopoiesis takes place in different waves and different locations throughout development, similar to mammals [24,25]. After emerging from the ventral dorsal aorta (VDA), hematopoietic stem and progenitor cells (HSPCs) travel to the primary site of embryonic hematopoiesis, the caudal hematopoietic tissue (CHT). Analogous to the mammalian fetal liver, the CHT is a vascular plexus in the ventral region of the tail between the caudal artery and cardinal vein [26]. From approximately 48 to 96 hours post fertilization (hpf) HSPCs exit circulation by transmigrating through the vascular endothelium and settle on the abluminal face of the vasculature. Here, local signals, including KIT ligand b, Oncostatin M, thrombopoietin, colony stimulating factor 3a, Chemokine (C-C Motif) Ligand 25b (Ccl25b), Chemokine (C-X-C Motif) Ligand 8b (Cxcl8b), and Chemokine (C-X-C Motif) Ligand 12a (Cxcl12a), regulate their trafficking and expansion [6,7,27]. Vascular endothelial cells remodel to form a pocket around resident HSPCs, and somite-derived stromal cells anchor HSPCs during CHT occupancy [28]. Initially, HSPCs populate more dorsal perivascular spaces in the CHT as sparse, individual cells. Over the course of about 2 days HSPCs expand and migrate more ventrally as the vasculature of the CHT becomes less complex. By 4 to 5 days post fertilization (dpf) clusters of HSPCs occupy a more ventral region of the CHT concentrated along the cardinal vein, and the number of HSPCs approximately doubles [29]. Primitive neutrophils regulate HSPC egress through secretion of Matrix Metallopeptidase 9 (Mmp9), which cleaves locally produced Cxcl12a and beginning at 4 dpf, HSPCs begin to seed the adult niche, the kidney, where the HSPCs reside, self-renew and differentiate to produce blood for the lifetime of the animal [27].

Since zebrafish exhibit orthologues for >82% of human disease related genes [30], and there is high conservation of genes and molecular signaling pathways that are involved in hematopoiesis between zebrafish and mammals [31,32], they are an attractive and powerful model organism to study diseases of the hematopoietic system. Still, no study described the analysis of LCs within the CHT and the assessment of interactions with surrounding niche components.

Here, we describe an orthotopic larval zebrafish xenotransplantation model to study leukemia in a complex microenvironment. In our system, LCs homed to the expected niche site, engrafted and proliferated. Furthermore, a distinct localization pattern was observed and close and persisting interactions of LCs with macrophages were monitored, along with low rates of attraction of zebrafish macrophages to the CHT and only very few phagocytosis events. By adding the temporal dimension, which allows assessing dynamic changes in different cell populations in realtime, we propose a new tool to study leukemia in a functional environment. This allows to discover niche mediated effects on malignant processes such as leukemia development, maintenance and relapse.

## Materials and methods

### Ethical statement

Written approval of the representatives was given for all the primary materials used in this study. The study was performed in accordance with the institutional ethical review board (written approval of the 'Ethikkommission der Medizinischen Fakultät der LMU München' with the numbers 19–495, 068–08 and 222–10) and with the Helsinki Declaration.

All animal trials were performed in accordance with the official committee on animal experimentation. According to German law (Tierschutz-Versuchstierverordnung–TierSch-VersV—§14), no specific approved animal protocol was necessary for the present study. Zebrafish larvae did not exceed an age of 5 days post fertilization and therefore, they do not take up nutrition by themselves.

### Cell preparation

The B-cell precursor acute lymphoblastic leukemia cell line NALM-6 was cultured in RPMI medium (Gibco, San Diego, CA, USA) supplemented with 10% FBS (Gibco, San Diego, CA, USA), without antibiotics or additional glutamine. PDX cells were freshly isolated from spleens of transplanted NOD.Cg-Prkds$^{scid}$Il2$^{rgtm1Wjl}$/SzJ (NSG) mice. For short-term *in vitro* culture, cells were cultured in StemSpan™ SFEM Medium (STEMCELL Technologies, Vancouver, Canada).

For cell proliferation analysis, cells were stained with 5 µM CFSE (Invitrogen™, Waltham, MA, USA) and resuspended in the respective medium for overnight cultivation. The next day, cells were transplanted. A small fraction of cells was kept *in vitro* for analysis.

### Zebrafish xenotransplantation

Fluorescently labeled LCs were injected using pre-pulled borosilicate glass capillaries (1B120F-4, World Precision Instruments, Sarasota County, FL, USA) using a microinjector (FemtoJet 4i, Eppendorf, Hamburg, Germany) under a stereomicroscope (Stemi 2000-C, Zeiss, Oberkochen, Germany). Two dpf embryos of the Tg(kdrl:mCherry), Tg(mpeg-1:mCherry-F), AB wt or Casper line were anesthetized with 750 µM Tricain (Sigma-Aldrich, St. Louis, MO, USA) and approximately 200–500 cells were injected into the duct of Cuvier. Injected embryos were transferred to E3 medium and kept at 36˚C.

## Flow cytometric analysis

At indicated time points post transplantation, larvae were euthanized and cut with a sharp scalpel in two parts right after the yolk sac extension. Tails were collected, pooled as indicated, resuspended in DPBS containing 50 μg/mL Liberase™ TM Research Grade (Sigma-Aldrich, St. Louis, MO, USA) and digested for 20–30 min at 37˚C. The suspension was filtered through a 30 μm mesh, washed once with DPBS and analyzed using a BD LSR Fortessa X20. *In vitro* cultivated cells were directly subjected to flow cytometric analysis.

## Repetitive *in vivo* imaging

Larvae were anesthetized (750 μM Tricain), placed on an agarose plate (0.5% agarose in E3-medium) and oriented laterally. Larvae were covered with sufficient E3-medium to keep them moist. For each larva, a bright field and fluorescent image was acquired with a fluorescent stereomicroscope (SteREO Discovery.V8, Zeiss, Oberkochen, Germany) using the Axiovision software (8x magnification), that the whole CHT was visible. After completion, larvae were transferred to fresh E3-medium and further maintained.

To monitor engraftment of LCs after transplantation, the corrected total CHT fluorescence (CTCF), which determines the level of fluorescence within the CHT, was calculated from fluorescence microscopy images as follows: The brightfield image was used to draw a region of interest (ROI) of the CHT area. This ROI was applied to the fluorescent image. Using the 'Measure' function of ImageJ, the *area* of the ROI and the *mean fluorescent intensity* (MFI) were determined. For each larva, the integrated density was calculated. As a background level, the mean integrated density of ten untransplanted larvae was used. Ultimately, the CTCF was calculated for each individual transplanted larva. The following formulas were used:

$$\text{Integrated Density } [RFU*\mu m^2] = \text{Mean Inensity CHT } [RFU]* \text{ CHT Area } [\mu m^2]$$

$$\text{CTCF } [RFU*\mu m^2] = (\text{Integrated Density }_{\text{Transplanted}} - \text{Integrated Density }_{\text{Untransplanted}}) [RFU*\mu m^2]$$

## High resolution *in vivo* time lapse imaging

At indicated time points post injection, zebrafish xenografts were anesthetized and screened according to the presence of LCs within the CHT using a fluorescent stereomicroscope (SteREO Discovery.V8, Zeiss, Oberkochen, Germany). Successfully transplanted xenografts were selected randomly and embedded in a glass bottom dish (4 Chamber, 35 mm, #1.5, Cellvis, Mountain View, CA, USA) using pre-heated (50˚C) 1.5% low melting agarose (MetaPhor™ Agarose, Lonza, Basel, Switzerland) in E3-medium (750 μM Tricain).

Embryos were observed with an Andor Dragonfly 500 spinning disc confocal system attached to a Nikon Ti2 microscope and equipped with an Andor iXon 888 life electron multiplying charge-coupled device (EMCCD) camera (Nikon ECLIPSE Ti2; Andor Dragonfly 500; iXon Life 888) within a breeding chamber pre-heated to 36˚C. Temperature was maintained with a full-enclosure incubator equipped with a thermocouple to monitor temperature close to the specimen (Okolab, Italy). The CHT of individual larvae were imaged with a 20x CFI Plan Apochromat NA 0.75 air objective (Nikon, Japan) and 16-bit time lapse movies were generated. The ImageJ software was used to z-project the acquired multidimensional images by using maximal intensity projection. For visualization purpose, contrast enhancement was performed to get 0.1–0.3% saturated pixels. Three-dimensional (3D) surface renderings were performed using Imaris software. For specific renderings, surfaces were visualized with transparency as indicated in the figure legend.

## Results

### Acute lymphoblastic leukemic cells (ALL-LCs) home to the CHT after transplantation

Xenotransplantation of ALL-LCs into zebrafish was performed at 48 hpf (hours post fertilization), by injecting 200–500 cells into the common cardinal vein (CCV) [33]. Transplanted larvae were raised at 36˚C, as this temperature allows larvae to develop normally and human cells to survive and proliferate [34]. The CHT, which is the proper hematopoietic niche at this stage of development, was monitored for three subsequent days using microscopic and flow cytometric methods, to study LCs in an orthotopic manner (Fig 1A).

Transgenic NALM-6- eGFP cells, which is a BCP-ALL relapse cell line, were transplanted. Larvae were anesthetized and the caudal parts were imaged using a fluorescence stereomicroscope to monitor LCs at one and three days post injection (dpi). The eGFP signal of NALM-6 cells was detected in transplanted larvae (Fig 1B), which was absent in untransplanted larvae (S1A Fig). NALM-6 cells homed to the CHT, where they resided during the three days of follow up time (Fig 1B).

### ALL cells engrafted and proliferated

The images of the transplanted larvae in Fig 1B showed an increase of the fluorescent signal within the marked niche area of the representative larva within two days. This could indicate that more LCs homed to the niche over time, but possibly also that cells proliferated. To better characterize LC engraftment, this process was analyzed in more detail. LCs within the CHT area were quantified over time in order to monitor colonization of the niche. Using non-invasive *in-vivo* microscopy allowed the analysis of the same animal at different time points. For ten transplanted larvae, the leukemic burden within the niche was determined by calculating the corrected total CHT fluorescence (CTCF) at one and three dpi. On average, the CTCF significantly increased by 68% within 48 hours (Fig 1C).

A flow cytometric approach was established to quantify LCs of transplanted larvae. To assess sensitivity of the assay, different numbers of NALM-6 cells were transplanted. For 50 larvae, the CTCF was calculated at one dpi. Larvae were divided into groups of ten (group A-E), according to the CTCF, prior to analyzing 10 larvae pooled as one sample, using flow cytometry (Fig 1D). Group A included larvae with the highest CTCFs (~3 x$10^9$ RFU*μm$^2$), while for group E larvae with the lowest CTCFs were pooled (~5 x$10^7$ RFU*μm$^2$) (Fig 1E). Representative dot plots of the flow cytometric analysis showed higher numbers of NALM-6 cells in group A than in group C (Fig 1F). As expected, for untransplanted larvae, no eGFP positive events were detected (S1B Fig). Furthermore, the correlation of imaging and flow cytometry for all five groups showed that both methods correlated linearly, as an increased fluorescent signal of the imaging analysis resulted in an increased number of detected cells as determined by flow cytometry (Fig 1G). Hence, both assays could equally be used to monitor engraftment of LCs at different time points.

To directly demonstrate LC proliferation *in-vivo*, a CFSE dilution assay was performed using the NALM-6 cell line and primary cells from two ALL patients (ALL-PDX: ALL-199 and ALL-265) (S1 Table). ALL-PDX cells expressing a fluorescent protein were generated by lentiviral transduction and serial passaging in NOD.Cg-Prkds$^{scid}$Il2$^{rgtm1Wjl}$/SzJ (NSG) mice [35]. Here, fluorescent protein expressing LCs (mCherry+ NALM-6; mTagBFP+ PDX cells) were labeled with the fluorescent proliferation sensitive dye CFSE, which covalently binds to proteins of the cells. Upon cell divisions, cells with halved CFSE signal can be detected using flow cytometry. Groups of transplanted larvae were sacrificed at one, two and three dpi and LCs

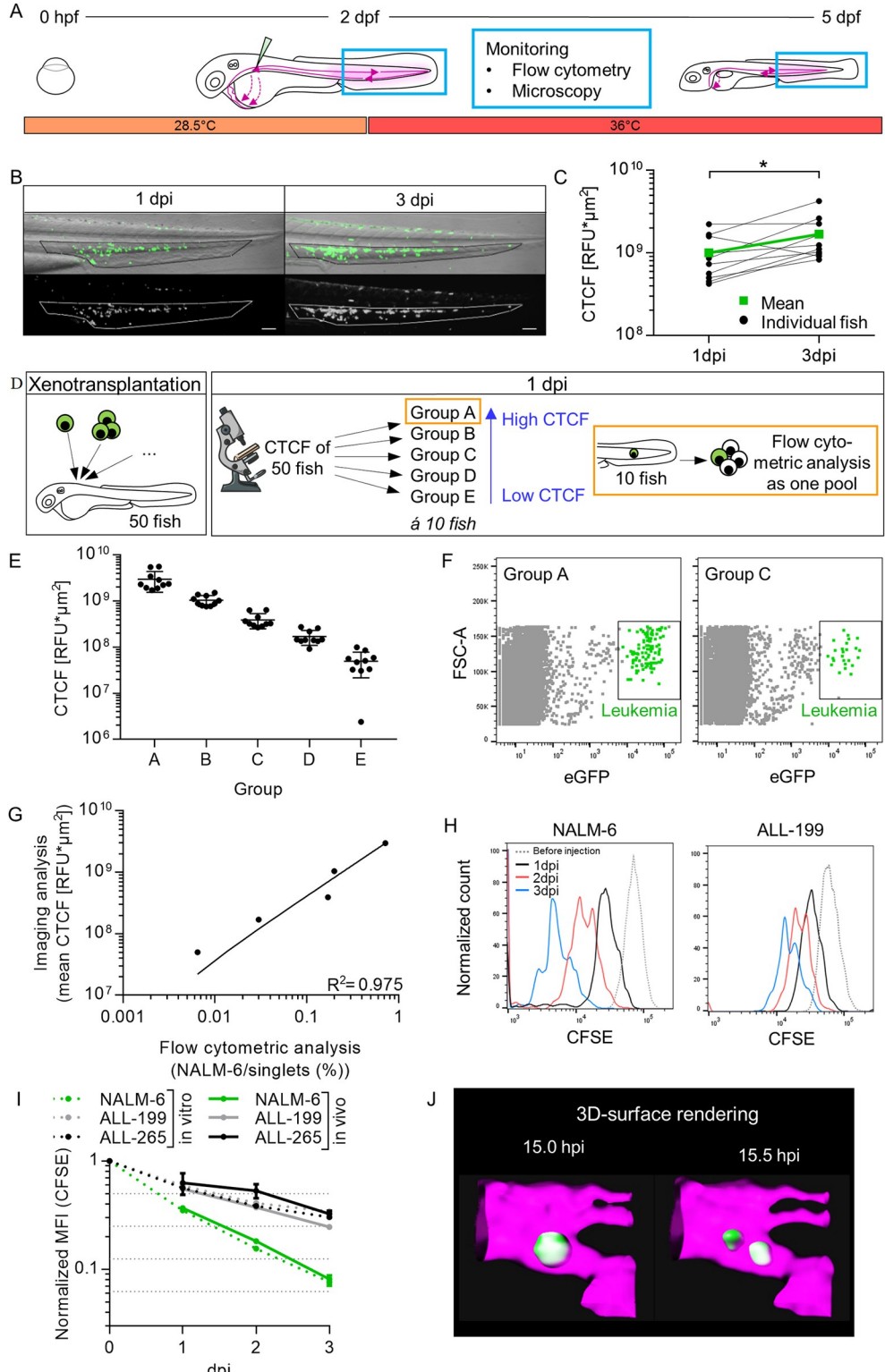

**Fig 1. Establishment of zebrafish larval xenograft of human ALL cell line and PDX cells. (A)** Schematic illustration of the workflow to monitor LCs within an orthotopic niche: Fertilized eggs were bred at 28.5˚C. At 2 dpf fluorescently labeled LCs got injected via the common cardinal vein. Subsequently, larvae were bred at 36˚C. In the following days, the CHT of larvae were analyzed using flow cytometry and microscopy. **(B)** Representative fluorescent (lower panel) and brightfield (merge, upper panel) images of the caudal part of a transplanted larva (NALM-6), at 1 and 3 dpi. The

outline of the niche area is highlighted and was used to assess the corrected total CHT fluorescence (CTCF) for transplanted larvae. **(C)** Measurement of the CTCF of 10 larvae at 1 and 3 dpi. Mean CTCF is indicated in green (n = 10, Wilcoxon matched-pairs signed rank test, * p≤0.05). **(D)** Experimental setup: Larvae were transplanted with NALM-6 cells (eGFP+). At 1 dpi CTCF was calculated and larvae were analyzed in groups of 10 according to the values obtained. Larvae of each group were pooled and measured as one sample using flow cytometry. **(E)** Grouping of the 50 analyzed larvae at 1 dpi with corresponding CTCF. Scatter-plot displays the calculated CTCF value per larva and mean value ± SEM per group. **(F)** Dot plot of the flow cytometry measurement of two representative transplanted samples (group A and C). Grey dots show cells in singlets gate; green dots show NALM-6 cells. For detailed gating see S2A and S2B Fig. **(G)** Correlation of the flow cytometric analysis and imaging analysis. One dot represents one group (for imaging analysis, mean value of the 10 larvae (E) is depicted, for flow cytometric analysis, the result of the pooled sample is depicted). Correlation curve and $R^2$ was calculated with linear regression. **(H)** Representative histograms of the CFSE signal of different transplanted LCs at daily measurements of 10 pooled larvae are depicted. **(I)** Quantification of the normalized MFI of the CFSE signal over time (*in vivo* samples: n = 3 (NALM-6: Mean value of three independent experiments performed in duplicates; ALL-199 & ALL-265: Values of one experiment performed in triplicates), mean value ± SEM; *in vitro* samples: NALM-6: n = 3 values of three independent experiments; ALL-199 & ALL-265: n = 1 value of one measurement in one experiment). **(J)** 3D- surface rendering of a NALM-6 cell undergoing cell division.

were identified using the fluorescent marker and the CFSE intensity of the cells was analyzed (for gating see S1C Fig). The histogram of NALM-6 and ALL-199 cells in Fig 1H showed the CFSE signal of all transplanted LCs at each timepoint. Importantly, defined populations with halved CFSE-signal appeared over time, representing cells that underwent cell division (Figs 1H and S1D). Quantifying the MFI of three different transplanted leukemia samples over time showed that NALM-6 cells proliferated faster than the two analyzed PDX cells (Fig 1I). ALL-199 and ALL-265 showed similar proliferation rates. Comparing the MFI of LCs *in vivo* and *in vitro* showed similar cell division kinetics. Interestingly, ALL-199 showed slightly increased proliferation under *in vivo* conditions when compared to *in vitro* conditions (Fig 1I). On average, xenografted PDX cells divided twice within three days (Fig 1I); however, histograms show that some cells underwent up to three cell divisions (Fig 1H). For NALM-6 cells, the number of average cell divisions was four (Fig 1H), while the maximum within three days, was five divisions (Fig 1I).

In addition, *in vivo* time lapse shots of confocal microscopy clearly confirmed a cell division of a NALM-6 cell within the vasculature of a zebrafish at 15.5 hours post injection (hpi). After cell division, the two daughter cells remained at the site of division (S1E Fig). 3D surface rendering confirmed the presence of only one cell at the beginning and excluded one cell being behind the other (Fig 1J).

Taken together, these data demonstrate the successful xenotransplantation of human LC lines and PDX cells into larval zebrafish with homing to the physiological niche and cell division.

## LCs and macrophages interacted for an extended period of time without phagocytosis

To further characterize LC interactions with the microenvironment, LCs and two different niche cell types were analyzed by time-lapse fluorescence microscopy. In a first step, the relative frequencies and distribution patterns of macrophages within the niche were examined for transplanted and untransplanted larvae. NALM-6 cells (eGFP+) were transplanted into Tg (mpeg1: mCherry-F) zebrafish larvae. At one dpi, the tail of transplanted larvae and untransplanted siblings were imaged using confocal microscopy. Macrophages were found throughout the tail but were mainly localized in the CHT. This was similar for transplanted and untransplanted larva (Fig 2A).

Time-lapse confocal microscopy was conducted for up to 10 hours to monitor interactions of LCs and macrophages over time. *Tg(mpeg1: mCherry-F)* larvae were transplanted with

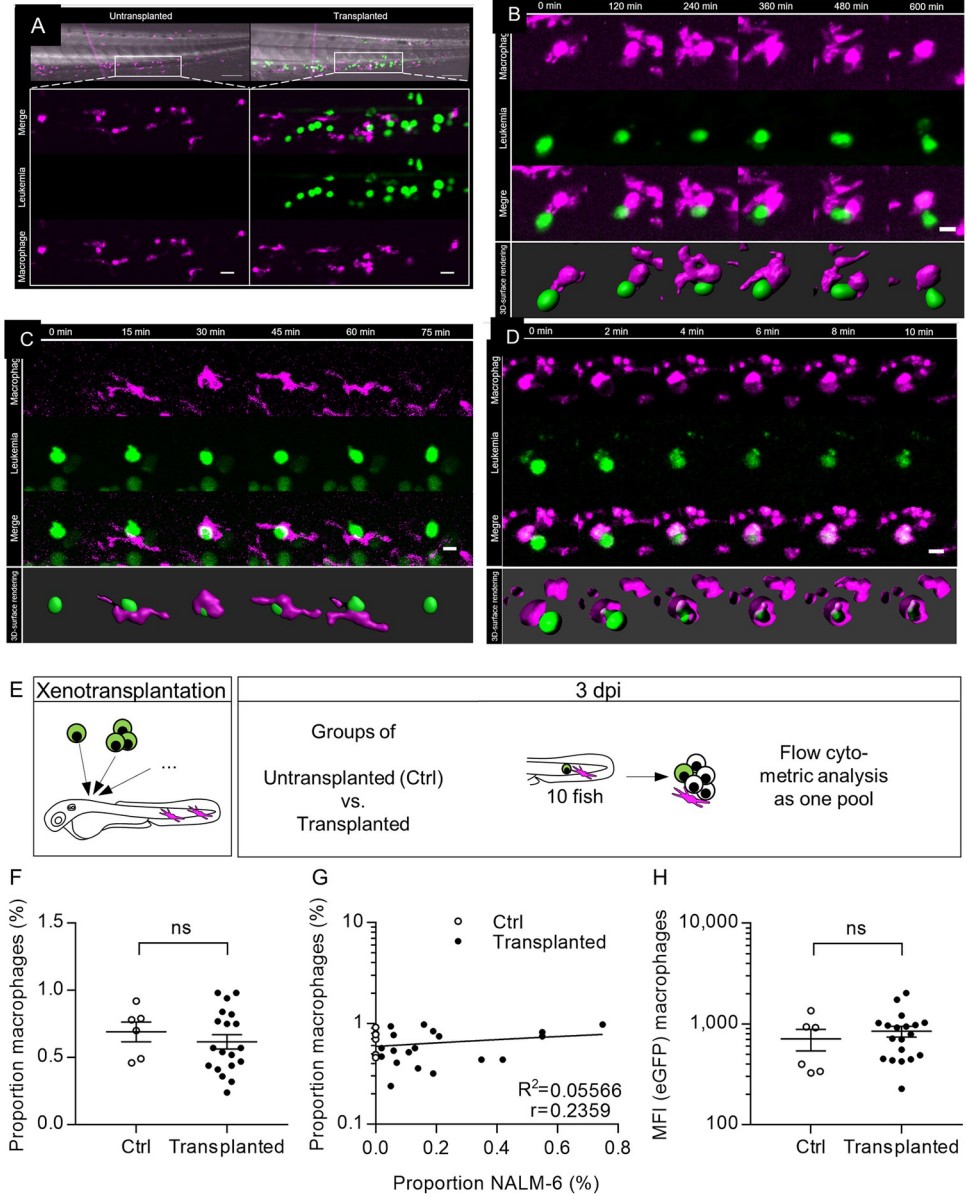

**Fig 2. Extended interaction between LCs and zebrafish macrophages. (A)** On the top, representative overview images of the caudal parts of an untransplanted (left) and a transplanted larva (right) at 1 dpi are depicted. Scale bar: 100 μm. Upper panels of the zoomed sections depict merged images of eGFP+ NALM-6 cells (green, middle row) and mpeg1:mCherry-F+ macrophages (magenta, lower row). Scale bar: 20 μm. **(B, C, D)** Six still frames of representative time lapse movies. Selected frames of the movies are depicted. Images show one macrophage mpeg1:mCherry-F+ (magenta, upper row) and LC (green, middle row) over time. Lower row shows the merged images. Scale bar: 10 μm. Separate panel below shows 3D-surface renderings. **(B)** Time lapse movie of a fish transplanted with NALM-6 cells (eGFP) starting at 31 hpi with a frame interval of 20 min. **(C)** Time lapse movie of a fish transplanted with NALM-6 cells (eGFP+) starting at 35 hpi with a frame interval of 5 min. **(D)** Time lapse movie of a fish transplanted with primary LCs (CFSE) starting at 23 hpi with a frame interval of 2 min, showing phagocytosis event. **(E)** Experimental setup: Tg(mpeg1:mCherry-F) larvae were transplanted with NALM-6 cells (eGFP+). At 3 dpi untransplanted (Ctrl) and transplanted larvae were grouped into groups of 10 and measured as one sample using flow cytometry **(F)** Quantification of the percentage of macrophages within singlets was assessed using flow cytometry. Each dot of the scatter-plot represents a group of ten larvae measured as one sample in three independent experiments. For gating details see S2 Fig **(G)** Dot plot of the correlation of the percentage of macrophages and NALM-6 cells of all samples analyzed in (F). Correlation curve was calculated using linear-regression (r = 0.2359, R2 = 0.056). **(H)** Quantification of the eGFP MFI of all detected macrophages using flow cytometry. Each point of the scatter-plot represents the MFI of all macrophages within the sample. **(F, H)** Mean value ± SEM is depicted. (untransplanted: n = 6; transplanted: n = 19). (unpaired t-test with Welch's correction, ns not significant).

NALM-6 cells (eGFP+). At indicated time points post transplantation, larvae were anesthetized, embedded and imaged. In Fig 2B, one representative interaction was documented, which lasted for more than 10 hours. At the start of the interaction (0 min) it was visible that the macrophage forms two pseudopodial extensions that reach towards a LC. At 240 min and 360 min, it is visible how the macrophage embraced the LC during the interaction. 3D-surface renderings support the observations (Fig 2B). In Fig 2C, a shorter lasting interaction is depicted. The LC was stationary during the interaction while the macrophage arrived rapidly and embraced the LC within 15 min. However, after 45 min the contact ended and the macrophage moved away from the LC.

As close interactions of LCs and macrophages were observed and macrophages occasionally embraced LCs, we asked whether zebrafish macrophages can phagocytize human LCs or not, due to inter-species incompatibilities. Time lapse movies of transplanted larvae were manually reviewed looking for green fluorescent material of a LC within the purple signal of macrophages which we assumed to be a phagocytosis event. In Fig 2D, one representative phagocytosis event of a LC by a macrophage is depicted in real-time. Two minutes after contact, the macrophage began to embrace the LC. Six minutes after close contact, the LC was completely phagocytized by the macrophage. In the 3D renderings of Fig 2D, surfaces of macrophages were depicted with high transparency to demonstrate enclosure of the LC within the macrophage (Fig 2D). Taken together, this demonstrated that LC elimination via macrophage phagocytosis is possible in general in our xenotransplant-setting.

## No enhanced recruitment of macrophages after xenotransplantation

After observing intensive contact between LCs and macrophages, we asked whether more macrophages were recruited to the CHT of transplanted larvae when compared to untransplanted control siblings. To investigate this, Tg(mpeg1: mCherry-F) larvae were transplanted with NALM-6 cells (eGFP+). Groups of ten larvae, either untransplanted (control (Ctrl)) or transplanted, were sacrificed at three dpi and the tails were analyzed by flow cytometry (Fig 2E). Proportions of macrophages among all singlets within the CHT were analyzed and compared between transplanted and control (Ctrl) larvae. The number of macrophages in larvae transplanted with LCs was similar to Ctrl larvae (Fig 2F). There was no correlation between leukemic burden and the number of macrophages identified in transplanted embryos (Fig 2G). In a next step, the MFI of all macrophages in the respective sample was calculated as a measure of phagocytosis activity. In case a macrophage engulfed a NALM-6 cell, the cells' fluorescence can be detected as demonstrated using confocal microscopy (Fig 2B and 2C). Here, no increased eGFP signal was detected for the macrophages of transplanted larvae when compared to Ctrl siblings (Fig 2H).

Taken together, macrophages were in contact with LCs as extended interactions were observed. In addition, zebrafish macrophages pursued their host-defense function as they phagocytized some malignant LCs.

## LCs left the vasculature and formed clusters

Endothelial cells interact with normal and malignant blood cells and they are a prominent cell type within the CHT. This tissue can generally be considered as a very well perfused sinusoidal vascular plexus delimited by the caudal artery (CA) and vein (CV) [27]. On consecutive days after transplantation of NALM-6 cells (eGFP+), Tg(kdrl:mCherry) larvae were anesthetized and the CHT was imaged and analyzed using confocal microscopy. At one dpi, LCs were evenly distributed throughout the niche (Fig 3A). At two dpi distinct LC clusters emerged that were located extravascular within the CHT, ventral of the CA. After transplantation, these

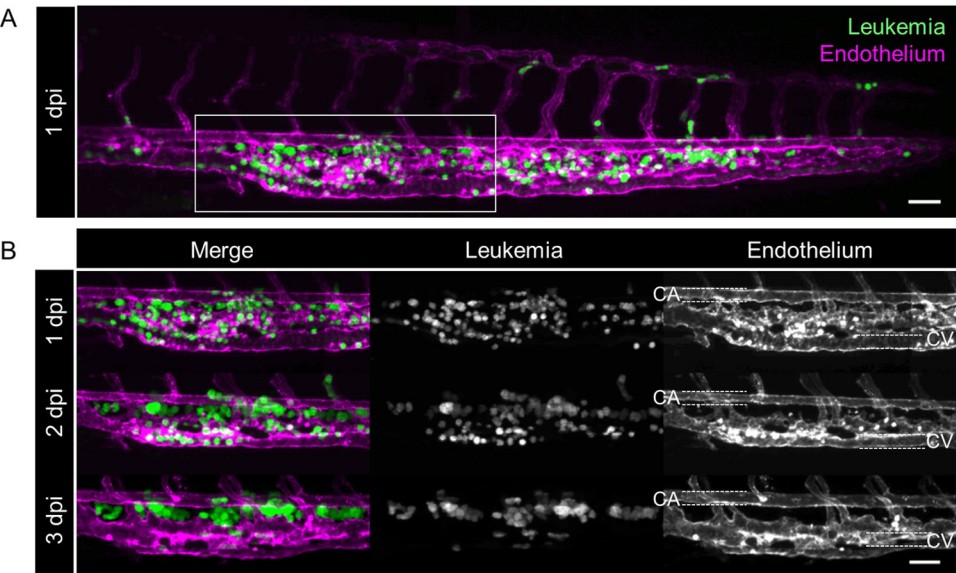

**Fig 3. Interaction between LCs and zebrafish endothelial cells.** Confocal microscopy of one representative transplanted Tg(kdrl:mCherry) larva over the course of three days is depicted. **(A)** Complete caudal part of the larva at 1 dpi. Marked area indicates field of view in (B). Scale bar: 50 μm. **(B)** Section of the CHT at three consecutive days. Left panel shows merged images of NALM-6 (green, middle panel) and the endothelium (kdrl:mCherry) (magenta, right panel). CV: Caudal vein (outlined with white dashed lines), CA: Caudal artery (outlined with white dashed lines). Scale bar: 50 μm.

pockets were colonized with many LCs that formed clusters. At three dpi, LCs were almost exclusively found extraluminal (Fig 3B). This observation demonstrated that LCs were able to extravasate the endothelium. Extravasation is a complex process, since it requires signaling between extravasating cells and the endothelium [36]. Strikingly, this specific behavior of LCs was not exclusively observed for NALM-6 cells (Fig 3), as ALL-199 and ALL-265 and some primary samples showed the same specific behavior (S3 Fig). This suggests a functional interaction between human leukemic cells and zebrafish endothelial cells.

## Discussion

The role of the microenvironment during BCP-ALL development, maintenance and relapse onset is not well understood. However, recent studies suggest a pivotal role of the leukemic niche in these processes [10,11]. Still, appropriate models that enable the analysis of LCs in an orthotopic niche are lacking and most studies investigate LCs independent from their surrounding tissue. Commonly, the analysis of LCs in the context of a functional microenvironment is performed with *in vitro* co-culture experiments where LCs are cultured in the presence of typically only one other cell type [37,38]. Mostly, the addition of expensive cytokines and growth factors is necessary to ensure cell viability and to mimic the niche to some extent [39]. *In vivo* mouse models enable analysis of LCs in the proper niche, but they are very costly and tedious and only a small number of animals can be analyzed per experiment [40]. Therefore, zebrafish is an emerging animal model in cancer research: (i) Studies show an adequate conservation of the zebrafish hematopoietic niche to study human LCs in a reasonable complex microenvironment, resembling human bone marrow [31,32], (ii) a large number of animals can be analyzed for statistically relevant numbers per experiment, and (iii) zebrafish are optically clear during development which enables *in vivo* imaging [32].

Here, we describe a model to study leukemia-niche interactions *in vivo* using a larval zebra-fish xenograft model of pediatric BCP-ALL. We used cell lines and PDX cells and showed engraftment of LCs within the CHT and distinct interactions with niche macrophages and endothelial cells.

Larval xenotransplantation models of cell lines and primary cells of different tumor entities were published in the past [13,14], including leukemias [19–21]. Current gold standard for the developmental stage for xenotransplantation is 48 hpf [33]. The growing number of studies published showed that the transplantation procedure and subsequent rearing temperature differed greatly, depending on the research question. Human cancer cell injections directly into the yolk sac were widely used for pre-clinical drug testing [20,21] and the analysis of the metastatic phenotype of solid tumors [41], while systemic injections into the blood circulation were mainly used to study cell dissemination [42] and metastatic features [15]. In the present study, LCs were transplanted into the blood via the duct of Cuvier and allowed to disseminate throughout the whole body by blood circulation. In order to monitor LC homing and engraftment within the orthotopic niche, dissemination of the transplanted LCs was evaluated using *in vivo* imaging, demonstrating homing of LCs to the hematopoietic niche, the CHT area.

Given the differential body temperatures of humans and fish of 37˚C and 28.5˚C, respectively, it is an ongoing debate how to compromise best between the temperatures to not harm fish development but to support cancer cell proliferation and engraftment. Up to a developmental stage of 48 hpf, rearing at the physiologic temperature of fish is very important for optimal development. Rearing of the embryos at 36˚C after 48 hpf showed only minor mortality rates or malformations of the larvae when compared to rearing starting earlier at 36˚C [43]. Subsequent to the transplantation, increasing the rearing temperature to 36˚C enhanced proliferation and survival of human cells greatly, when compared to 34˚C [34], which was widely used in other studies [16,19], while transcriptional differences remained minor [43]. Adequate development of the larvae upon transplantation was essential when analyzing niche components of the fish; however, engraftment of LCs was considered equally important for the intended model. In conclusion, for the present project, embryos were bred at 28.5˚C until 48 hpf [34]. Subsequent to the xenotransplantation of LCs, larvae were kept at 36˚C. In our experiments, no severe developmental abnormalities were observed and most of the larvae developed normally. This is in line with the results of Cabezas-Sainz et al. [34].

In our described orthotopic model, we observed LCs homing and engraftment in the CHT within three days of follow up, with the tumor burden increasing over time as monitored with serial *in vivo* imaging. Notably, not exclusively LCs are documented to home to the CHT. Sacco *et al.* transplanted multiple myeloma cells into the blood circulation of larvae and showed that the myeloma cells resided in the CHT area shortly after transplantation, while cells of a cervix carcinoma did not [42]. Interestingly, a breast cancer cell line, known to build bone marrow metastasis, also homed to the CHT area [42,44]. In addition, primary cells of a breast cancer bone marrow metastasis engrafted within the CHT area [44], supporting the idea of an appropriate conservation of zebrafish niche components to resemble a bone marrow like environment, recognized by human cancer cells.

When analyzing LCs after systemic transplantation, monitoring of the tumor burden at a single time point is not sufficient to clearly demonstrate whether LCs underwent cell divisions or if they homed more efficiently to the CHT over time. As demonstrated using *in vivo* time lapse imaging, our results clearly demonstrate cell divisions of human LCs within the zebrafish. Furthermore, the analysis with a proliferation sensitive dye using flow cytometry showed that individual cells underwent up to five cell divisions within only three days. This result is comparable to the data of Vargas-Patron *et al.*, who demonstrated up to four cell divisions of

glioblastoma cells within three days using a similar approach [45]. Notably, to our knowledge no other study demonstrated cell divisions within larval xenografts of LCs in such detail.

Yet, larval xenografts were mainly used to study drug response [20,21,46] or metastatic properties of solid tumors [15,16,41,44] and the model was seldom used to analyze interactions with surrounding tissues. In order to assess the suitability of the model to identify characteristics of the microenvironment that are important in processes such as tumor development, maintenance and relapse onset, we investigated if and how LCs were in contact with certain niche cells. Larval zebrafish are especially good recipients for xenotransplanted cells as they did not develop adaptive immunity yet [47]. Still, the analysis of cellular parts of the innate immune system is feasible, as macrophages are already present as early as 24 hpf [48]. To date, only very few studies exist that analyzed the interplay of macrophages and BCP-ALL. Thus, we assessed the feasibility to use larval zebrafish as a model to study macrophages in more detail.

Macrophages belong to cellular elements of the innate immune system and are highly interesting niche cells to study as they are known to play diverse roles in malignant diseases. Different activation states of macrophages are described: an immune-stimulating and therefore anti-tumor (M1) and an immune-suppressive and therefore pro-tumor (M2) state [49]. In zebrafish embryos, macrophages are present as early as 24 hpf [48] and can be easily monitored in transgenic larvae expressing the fluorescent protein mCherry specifically in macrophages (Tg (mpeg1:mCherry-F)). Using *in vivo* imaging of macrophage-transgenic reporter larvae, we were able to monitor macrophage behavior of zebrafish larvae after xenotransplantation of human leukemic cells. While fulfilling their host defending function, they patrol throughout the fish. Interestingly, some macrophages slowed down and were very stationary during the observed contact with LCs. We were able to document dynamic, close and persisting cellular interactions between LCs and macrophages, which lasted up to 10 hours. Especially 3D renderings emphasized the close contact of the two cells. At the onset of the contact, macrophages changed their shape and developed protrusions to reach the LC. Later, the LC was fully embraced by the macrophage before they released each other and moved on. Contrary to these observations, we saw rapid phagocytosis events of macrophages eliminating transplanted LCs. This demonstrated that macrophages showed a very diverse behavior towards transplanted LCs, as they were temporarily closely interacting with LCs while it was shown that they clearly were able to eliminate LCs. During these periods of close contact, macrophages and LCs could possibly communicate extensively, resulting in processes, such as niche remodeling and leukemia progression. For instance, Roh-Johnson *et al.* were able to demonstrate in melanoma zebrafish xenografts that during such close contacts macrophages transferred cytoplasm to the melanoma cell, which then led to enhanced melanoma dissemination [50]. In contrast, Asokan *et al.* documented a prolonged interaction of an immune cell with a breast cancer cell, which finally got phagocytized [51], demonstrating the versatile polarization potential of macrophages. In another recent publication, Póvoa *et al.* identified that a colorectal cancer cell population (progressors) was able to block macrophage-mediated tumor clearance of another colorectal cancer cell population (regressors) in a co-transplantation approach. Using individual recipients for both populations, the regressive population could not engraft, demonstrating the progressors' potential to modulate macrophages towards a pro-tumoral polarization [52].

Despite the observation of intensive contact between LCs and macrophages, no enhanced infiltration or even recruitment of macrophages to the caudal part of the larva could be observed at three dpi, when analyzing relative cell numbers in a flow cytometric approach. In addition, no extensive phagocytosis of the fluorescently labeled LCs was detected when analyzing the inclusion of LCs' fluorescence within macrophages. Still, sporadic phagocytosis events were observed and demonstrated the inter-species ability of phagocytes to eliminate human cells. In contrast to our observation of mild phagocytosis activity, Asokan *et al.* detected

significantly more macrophages co-localizing with breast cancer cells, as a measurement of phagocytosis activity, after a comparable period of time after transplantation as we used for our experiments [51]. This indicates differential behavior of macrophages within different tumor microenvironments.

Our data suggests a leukemia-supporting polarization of the macrophages, as a result of the close and persisting interactions, rather than an anti-tumor phenotype, with extensive phagocytosis and active macrophage recruitment to the niche. To our knowledge, this is the first study visualizing and analyzing the interaction of innate immune cells and LCs in a zebrafish model and demonstrates the potential of the model to study their interplay in more detail.

In a next step, the localization pattern of LCs within the CHT area over time was analyzed. The larval zebrafish model enables repetitive imaging of individual larvae at different days. Here, we were able to monitor an increase in LC burden over time. With regards to the localization of LCs, it became apparent that at one dpi, LCs were rather uniformly spread within the vasculature of the niche, while starting at two dpi, LCs tended to build clusters within physiologically emerging extraluminal pockets. At three dpi, LCs were almost exclusively localized within these extravasal pockets around the main artery. As we observed this dissemination pattern of LC localization for a number of different LCs, the extravasation and cluster formation seemed to be a specific process rather than a random behavior. Extravasation is a complex process and requires functional signaling between extravasating LCs and the endothelium [36], demonstrating a functional signaling between human LCs and the zebrafish endothelium in the present model. Subsequent to LC homing to the CHT, LCs were surrounded by a dense vessel bed, enabling signaling and thereby reprogramming of this niche component. For HSPCs, Tamplin *et al*. showed that the remodeling of endothelial cells around HSPCs, which they called "endothelial cuddling", led to HSPC proliferation [28]. A similar important function of the endothelium might be conceivable for LCs. The accumulation of cancer cells within extraluminal pockets was also described by Marcatali *et al*. when they were transplanting breast cancer cells. They observed these cells within these pockets for two more days, up to five dpi [44]. Interestingly, ventral of the main artery, at around three to five dpf, a vessel of the lymphoid system was emerging [53]. This is exactly the time span in which we observed the specific distribution pattern towards the pockets. Lymphatic organs are known to be infiltrated with tumor cells [54], so there is a possibility that the pockets are associated to lymphatic tissue. This hypothesis could be proven with a suitable reporter fish line. Notably, these extraluminal pockets are not colonized by healthy HSPCs. While HSPCs are also evenly distributed throughout the CHT at three dpf, they shift towards the caudal vein at four dpf, before relocation to the kidney marrow [29]. Therefore, it could also be speculated that oxygen tension or expression of different cytokine receptor types between leukemic cells and endogenous HSPCs could be responsible for these diverging migration patterns within the hematopoietic niche.

## Conclusions

Taken together, the generated larval xenograft model is suitable to study leukemia in a complex environment. In line with studies of other tumor entities [44,51,52], LCs showed close and persisting interactions with macrophages and a specific dissemination pattern within the vasculature, suggesting a functional interplay with the microenvironment. Thus, this model is useful to study effects of the leukemia microenvironment and will help to better understand its role in processes such as leukemia development, maintenance and relapse in the future.

## Supporting information

**S1 Fig. EGFP and CFSE intensities of NALM-6 and PDX cells. (A)** Representative fluorescent (lower panel) and brightfield (merge, upper panel) images of the caudal part of an untransplanted larva at 1 and 3 dpi. The outline of the niche area is highlighted and was used to assess the CTCF of the larvae. **(B)** Dot plot of the flow cytometry measurement of an untransplanted sample. Grey dots show cells in singlets gate; green dots show NALM-6 cells. For detailed gating see S2A and S2B Fig. **(C)** Gating strategy for single cell suspension of transplanted larvae: 'Lymphocyte' gate was established using FSC-A/SSC-A of a pure leukemia sample (not shown). This gate was applied on the target sample and processed further. 'Singlets' were gated using FSC-H/FSC-W. mCherry positive 'Leukemia' cells (here NALM-6) were selected by gating on mCherry positive events using mCherry/FSC-A. These cells were used to analyze CFSE intensities and distribution patterns. **(D)** Representative histograms of the CFSE signal of PDX-265 at daily measurements of 10 pooled larvae. **(E)** Six frames of a time lapse video between 14 and 16.5 hpi with a frame interval of 30 min are depicted. Upper row shows the merged image of eGFP+ NALM-6 cells (green, middle row) and kdrl:mCherry endothelial cells (magenta, lower row). These images illustrate a NALM-6 cell located in a vessel for an extended period of time. Between 15 and 15.5 hpi the NALM-6 cell divided. CA: Caudal artery (outlined with white dashed lines). Scale bar: 10 μm.
(TIF)

**S2 Fig. Detailed gating strategy for FCAS analysis of leukemic cells after xenotransplantation. (A)** 'Cells' were selected using FSC-A/SSC-A. 'Singlets' were gated using FSC-H/FSC-W. mCherry positive 'macrophages' were selected by gating on mCherry positive events using mCherry/FSC-A. 'Leukemia cells' were detected by applying the 'lymphocytes gate that was drawn using a leukemia cell sample (see (B)) on the singlets gate (FSC-A/SSC-A). Next, events that were positive for the fluorescent marker (here: eGFP) were considered as leukemia cells by eGFP/FSC-A. **(B)** Exemplary gates of the pure LC sample. **(C)** Exemplary plots of untransplanted larvae sample.
(TIF)

**S3 Fig. Imaging of PDX leukemic cells after transplantation.** Confocal microscopy of the caudal parts of either one representative Tg(kdrl:mCherry) larva, transplanted with PDX-ALL-199 cells (top) or PDX-ALL-265 cells over the course of three days is depicted. Left panel shows merged images of LCs (green, middle panel) and the endothelium (kdrl:mCherry) (red, right panel). CV: Caudal vein (outlined with white dashed lines), CA: Caudal artery (outlined with white dashed lines). Scale bar 50 μm.
(TIF)

**S1 Table. Clinical data of patients with BCP-ALL.**
(TIF)

## Acknowledgments

We thank Bettina Schmid and her team (German Center for Neurodegenerative Diseases (DZNE, Munich, Germany) for excellent animal care services. Christina Zeller for providing the eGFP lentivirus, Anna-Katharina Wirth for providing ALL-199 and ALL-265.

## Author Contributions

**Conceptualization:** Anja Arner, Bradley Wayne Blaser, Vera Binder-Blaser.

**Data curation:** Anja Arner, Andreas Ettinger.

**Formal analysis:** Anja Arner, Andreas Ettinger, Bradley Wayne Blaser.

**Funding acquisition:** Vera Binder-Blaser.

**Investigation:** Anja Arner.

**Methodology:** Anja Arner, Vera Binder-Blaser.

**Project administration:** Bettina Schmid, Vera Binder-Blaser.

**Resources:** Bettina Schmid, Irmela Jeremias, Vera Binder-Blaser.

**Software:** Vera Binder-Blaser.

**Supervision:** Bettina Schmid, Irmela Jeremias, Vera Binder-Blaser.

**Validation:** Anja Arner, Bradley Wayne Blaser, Vera Binder-Blaser.

**Visualization:** Anja Arner.

**Writing – original draft:** Anja Arner, Vera Binder-Blaser.

**Writing – review & editing:** Anja Arner, Andreas Ettinger, Bradley Wayne Blaser, Bettina Schmid, Irmela Jeremias, Nadia Rostam, Vera Binder-Blaser.

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
