## [Decision Letter · Decision Letter 0]

7 May 2024

PONE-D-24-13983In vivo monitoring of leukemia-niche interactions in a zebrafish xenograft modelPLOS ONE

Dear Dr. Binder-Blaser,

Thank you for submitting your manuscript to PLOS ONE. After careful consideration, we feel that it has merit but does not fully meet PLOS ONE’s publication criteria as it currently stands. Therefore, we invite you to submit a revised version of the manuscript that addresses the points raised during the review process.

We look forward to receiving your revised manuscript.

Kind regards,

Persio Dello Sbarba, MD, PhD

Academic Editor

PLOS ONE

Journal Requirements:

4. We note that Figure S1 in your submission contain copyrighted image. All PLOS content is published under the Creative Commons Attribution License (CC BY 4.0), which means that the manuscript, images, and Supporting Information files will be freely available online, and any third party is permitted to access, download, copy, distribute, and use these materials in any way, even commercially, with proper attribution. For more information, see our copyright guidelines: http://journals.plos.org/plosone/s/licenses-and-copyright.

a. You may seek permission from the original copyright holder of Figure S1 to publish the content specifically under the CC BY 4.0 license. 

5. We notice that your supplementary figures are uploaded with the file type 'Figure'. Please amend the file type to 'Supporting Information'. Please ensure that each Supporting Information file has a legend listed in the manuscript after the references list.

Additional Editor Comments:

Dear Dr. Binder-Blaser,

your paper received positive comments and will be suitable for publication provided you answer the points the reviewers raised.

persio dello sbarba

Reviewers' comments:

Reviewer's Responses to Questions

**Comments to the Author**

1. Is the manuscript technically sound, and do the data support the conclusions?

Reviewer #1: Yes

Reviewer #2: Yes

2. Has the statistical analysis been performed appropriately and rigorously? 

Reviewer #1: Yes

Reviewer #2: Yes

3. Have the authors made all data underlying the findings in their manuscript fully available?

Reviewer #1: Yes

Reviewer #2: Yes

4. Is the manuscript presented in an intelligible fashion and written in standard English?

Reviewer #1: Yes

Reviewer #2: Yes

5. Review Comments to the Author

Reviewer #1: The authors present the manuscript entitled In vivo monitoring of leukaemia-niche interactions in a zebrafish xenograft model.

Briefly, the manuscript is well written but some sections must be clarified. You will find all my requests attached.

Reviewer #2: The authors describe a zebrafish model to quantitatively analyze leukemic cells (LCs) in their microenvironment, to study niche-mediated leukemogenesis, leukemia maintenance and relapse. Zebrafish is an actractive model for studying hematopoiesis, hematopoietic niche and cancer. Indeed cell-cell interactions in zebrafish are hightly conserved in mammals. Moreover, the proposed model is useful to investigate ALL considering LCs not indipendently from their microenvironment, but in their niche.

The article is well written, the used procedures well explained and the figures well comprehensibles.

A few clarifications would be useful: since the bone marrow niche is characterized by peculiar conditions, such as hypoxia, have you measured the oxygen tension and the pH in CHT? and what about the expression of different niche factors in transplanted/untransplanted larveae?

6. PLOS authors have the option to publish the peer review history of their article (what does this mean?). If published, this will include your full peer review and any attached files.

Reviewer #1: No

Reviewer #2: No

---

## [Author Response · Author response to Decision Letter 0]

12 Aug 2024

Overall Response: We wish to thank the reviewers for their thoughtful comments on our manuscript. We have addressed all of the reviewers’ comments below and as indicated in yellow in the text of the revised manuscript. We believe the revised manuscript is substantially improved as a result of these changes. 

All the changes that are not highlighted in yellow are due to the formal ‘Journal Requirements’ that were not met in the first submitted manuscript. 

 (All the new line numbers listed below are referring to the manuscript without tracked changes and the manuscript with tracked changes set to “Simple Markup”. When set to “All Markup”, the line numbers change. We just wanted to make the reviewers aware of that.)

Reviewer 1 comments:

It is a very promising tool and I hope you will be able to investigate even more on this because it might provide, in next future, a concrete way to understand the interactions between cancer stem cells and their niche and on how both promotes the changes in the counterpart.

Briefly:

Abstract: brief but straight to the point.

Line 30: comma after “transplant”.

Response: Thank you for the comment, changing the commas made the content more clear (line 30). 

Introduction: well written but poor from the BCP-ALL side. I would like you to describe and widen the leukaemia part. 

Response: We have expanded on BCP-ALL in the opening paragraph of the introduction (lines 38-45), addressing some of the background of the disease and the survival rates for initial disease and relapse. This leads over to why also looking at the microenvironment in ALL could be important to prevent relapse. 

Line 55-58: the whole period is very complex and not well written, please re-write it.

Response: We have rewritten this paragraph (now lines 56-60) by deleting the complex part of “temporal dimension” and added this at the end of the introduction (now lines 101-103) to clarify what makes the model interesting.

Mat and Methods: very well written. The authors have meticulously described all the tools, techniques, materials and reagents used in the study.

Line 96: please provide supplements of RPMI 1640 medium i.e. pen/strep, glutamine addition.

Response: Nalm-6 cells were cultivated without antibiotics, and without supplemental glutamine. This information has been added to the revised manuscript (now lines 120-121). 

Line 125: comma after “)”. 

Line 126: comma after “CHT”. 

Line 129: change was with were determined. 

Response: These changes have been made (now lines 149, 150 and 153). 

Results: well written even though some points must be cleared up. 

Line 159-161: not easily comprehensible. Please, re-write the sentence.

Line 171: proliferate-d. 

Response: We re-wrote the sentence and hope that it is easier to comprehend for the reader (now lines 185-187). Thank you for pointing out the grammar mistake (now line 197). 

Discussion: This part is well structured and substantiates authors' findings. 

Line 345-347: re-write the sentence in a clearer way.

Response: We have rewritten the sentence and added 2 additional sentences to make the advantages of the zebrafish as an animal model in cancer research more clear (now lines 376-383).

Considerations and questions to be answered (does not follow the order of the presented results):

Leukaemic Stem Cells (LSCs) reside within the leukaemic stem cell niche where their fate is decided. It is also well known that, within an heterogenous population, only a very small percentage comprehends leukaemic stem cells. In your latest part of the results, you show that almost the entirety of the injected LCs are capable to extravasate and colonize the niche. This should mean that you have a very rich stem population at time 0 of injection. 

Is it true? 

Response: Indeed, we observed that almost all the cells we transplanted did extravasate to the hematopoietic niche. We assume that this happens due to the cytokine milieu created by the hematopoietic niche (now summarized in an additional paragraph in the introduction (lines 72-91), as it is true for endogenous hematopoietic stem and progenitor cells (HSPCs). However, we are not able to make any claims about the stemness of these particular cells. Leukemia stem cells are not well-defined in ALL which contrasts with AML. In normal hematopoietic cell transplants and in AML, non-stem cell populations are able to home and engraft in the hematopoietic niche, so the observation that most cells in this study are extravascular does not necessarily speak to their stemness. In order to show “leukemic stemness”, one would have to show subsequent regrowth within the niche over a prolonged period. This type of experiment is out of the scope of what can be achieved in the embryo model. In summary, we saw almost all of the LCs extravasate, but we don’t know how many of these cells have stem cell potential. 

Do you have any idea why almost all of your cells extravasate between day 2 and 3 post injection and almost none remains in the vessel?

Response: We speculate that they follow the route of normal hematopoietic stem cells in the hematopoietic niche, as it is now summarized in the introduction (lines 72-91, labelled in yellow), guided by the same cytokines as endogenous hematopoietic stem and progenitor cells. We further speculate, that cells that do not achieve this, fail to receive local growth factor support in the niche and do not survive.

Could you provide proof or idea of how NALM-6 cell line changes when it extravasates (i.e. expression of different adhesion markers for example, different metabolism)? It could be interesting to understand if, when extravasated, cells population enriches in terms of stemness.

Response: This is a very interesting point. We are planning to analyse this by single cell RNA sequencing of the transplanted cells at different time points after transplantation. We believe that this is outside the scope of the current study and will be the subject of another manuscript.

Have you gone further 3 days post injection to verify what happens to your cells in the CHT? Do you think it could be feasible? Do you think they might stay there or maybe colonize other parts of the animal?

Response: We are currently planning these experiments for a subsequent manuscript. We would expect them to migrate further to the kidney, the adult site of hematopoiesis in the fish (now summarized in the introduction, lines 72-91, labelled in yellow).

Could you describe a little bit further, in the intro section or in the discussion, the CHT part of the zebrafish and how it might interact and sustain ALL homing? You often speak about the niche but you never describe it very well. 

Response: We added a paragraph on the hematopoietic niches throughout development and how the different cell types and cytokines they produce influence migration of endogenous HSPCs (Introduction, line 72-91, labelled in yellow). We speculate that the same cytokines influence the migration of the transplanted leukemic cell line.

It could be useful for the reads if you indicate with a text the position of the vessels in the last images you presented.

Response: These changes have been made to Figure 3 and Supplementary Figure S1 und S3. An explanatory sentence was added to the respective figure legends (lines 365-366, 737-738, 752-753). 

Reviewer 2 comments: 

The authors describe a zebrafish model to quantitatively analyze leukemic cells (LCs) in their microenvironment, to study niche-mediated leukemogenesis, leukemia maintenance and relapse. Zebrafish is an actractive model for studying hematopoiesis, hematopoietic niche and cancer. Indeed cell-cell interactions in zebrafish are hightly conserved in mammals. Moreover, the proposed model is useful to investigate ALL considering LCs not indipendently from their microenvironment, but in their niche.

The article is well written, the used procedures well explained and the figures well comprehensibles.

A few clarifications would be useful: since the bone marrow niche is characterized by peculiar conditions, such as hypoxia, have you measured the oxygen tension and the pH in CHT? and what about the expression of different niche factors in transplanted/untransplanted larveae?

Response: These are very important questions, but we did not measure oxygen tension or the pH throughout the CHT. We could only speculate, that the transplanted leukemic cells might be driven by oxygen differences, when, at 3 days post transplantation, they migrate towards the caudal artery, in the very opposite way of the endogenous HSPCs, that migrate towards the caudal vein. We added an additional sentence at the end of the Discussion (lines 513-515) and hope that together with the additional paragraph on the hematopoietic niche and niche factors in zebrafish (line 72-91), and the highlighted (not changed) sentence in line 421-423, this draws a more clear picture of our hypotheses regarding our observations. 

The question of expression changes of niche factors is a very important one, together with the question Reviewer 1 raised about the expression changes of the leukemic cells after extravasation. We are pursuing answers to these questions using single cell RNA sequencing in transplanted and untransplanted larvae and aim to report these in a future manuscript.

---

## [Editor Report · Decision Letter 1]

13 Aug 2024

In vivo monitoring of leukemia-niche interactions in a zebrafish xenograft model

PONE-D-24-13983R1

Dear Dr. Vinder-Blaser,

We’re pleased to inform you that your manuscript has been judged scientifically suitable for publication.

We apologize for the delays you encountered in the course of submission.

Within one week, you’ll receive an e-mail detailing the required amendments necessary in view of formal acceptance for publication. When these have been addressed, you’ll receive a formal acceptance letter and your manuscript will be scheduled for publication.

Kind regards,

Persio Dello Sbarba, MD, PhD

Academic Editor

PLOS ONE
---

## [Editor Report · Acceptance letter]

22 Aug 2024

PONE-D-24-13983R1 

PLOS ONE

Dear Dr. Binder-Blaser, 

I'm pleased to inform you that your manuscript has been deemed suitable for publication in PLOS ONE. Congratulations! Your manuscript is now being handed over to our production team.

Kind regards, 

on behalf of

Prof Persio Dello Sbarba 

Academic Editor

PLOS ONE